# Efficacy, Safety, and Concerns on Microbiota Modulation, Antibiotics, Probiotics, and Fecal Microbial Transplant for Inflammatory Bowel Disease and Other Gastrointestinal Conditions: Results from an International Survey

**DOI:** 10.3390/microorganisms11112806

**Published:** 2023-11-19

**Authors:** Tommaso Lorenzo Parigi, Sophie Vieujean, Kristine Paridaens, Kira Dalgaard, Laurent Peyrin-Biroulet, Silvio Danese

**Affiliations:** 1Department of Gastroenterology and Digestive Endoscopy, IRCCS Ospedale San Raffaele, 20132 Milan, Italy; parigi.tommaso@hsr.it; 2Division of Immunology, Transplantation and Infectious Disease, Università Vita Salute San Raffele, 20132 Milan, Italy; 3Hepato-Gastroenterology and Digestive Oncology, University Hospital CHU of Liège, 4000 Liège, Belgium; 4Ferring International Center S.A., CH-1162 Saint-Prex, Switzerland; kristine.paridaens@ferring.com; 5Ferring Pharmaceuticals A/S, 2770 Kastrup, Denmark; kira.dalgaard@ferring.com; 6Department of Gastroenterology, Nancy University Hospital, F-54500 Vandœuvre-lès-Nancy, France; 7INSERM, NGERE, University of Lorraine, F-54000 Nancy, France; 8INFINY Institute, Nancy University Hospital, F-54500 Vandœuvre-lès-Nancy, France; 9FHU-CURE, Nancy University Hospital, F-54500 Vandœuvre-lès-Nancy, France; 10Groupe Hospitalier Privé Ambroise Paré–Hartmann, Paris IBD Center, F-92200 Neuilly sur Seine, France; 11Division of Gastroenterology and Hepatology, McGill University Health Centre, Montreal, QC H4A3J1, Canada

**Keywords:** microbiota, antibiotics, probiotics, fecal microbial transplant, inflammatory bowel disease, Crohn’s disease, ulcerative colitis, pouchitis, irritable bowel syndrome

## Abstract

The gut microbiota play a pivotal role in human health. Dysbiosis, alterations in microbiota composition and function, is associated with gastrointestinal disorders, including inflammatory bowel disease (IBD). This international survey aimed to assess physicians’ experiences, perceptions, and practices related to microbiome modulation for gastrointestinal conditions, with a focus on IBD. Results from 142 healthcare professionals, predominantly gastroenterologists, confirmed a consensus on the relevance of the gut microbiota in IBD pathogenesis. However, the utilization of microbial composition analysis and probiotics in clinical practice was limited, primarily due to the lack of standardized guidelines and supporting evidence. Physicians held conflicting views on antibiotics, recognizing their potential for inducing remission but also causing flares in IBD. Respondents also had varying opinions on the efficacy of fecal microbiota transplantation (FMT) for different gastrointestinal conditions, with higher confidence in FMT effectiveness for irritable bowel syndrome with diarrhea, pouchitis, and ulcerative colitis. Concerns on FMT included uncertainty about effect duration, administration intervals, and conflicting evidence. Donor selection was believed to be a crucial factor in FMT outcomes. This survey highlights the need for further research and evidence-based guidelines to optimize the use of microbiome-based therapies in clinical practice. As our understanding of the gut microbiome continues to evolve, these insights will contribute to more informed and personalized approaches to managing gastrointestinal disorders.

## 1. Introduction

The gut microbiota are a complex and dynamic community of microorganisms that colonizes the human gastrointestinal tract and plays an important role in various aspects of host physiology and health. The gut microbiota are involved in the digestion and absorption of nutrients, the synthesis of vitamins and metabolites, the modulation of the immune system, and the protection against pathogens. However, the gut microbiota can also be a source of inflammation and disease when its composition and function are altered. This phenomenon, known as dysbiosis, has been associated with nearly all gastrointestinal disorders and most prominently inflammatory bowel disease (IBD) [1], irritable bowel syndrome (IBS), colorectal cancer [2,3,4], and celiac disease [5].

IBD is a chronic and relapsing inflammatory disorder of the intestine that comprises two main subtypes: ulcerative colitis (UC) and Crohn’s disease (CD) [6,7]. IBD affects millions of people worldwide, with a significant impact on quality of life and an increasing burden on healthcare systems [8]. The precise etiology of IBD remains unclear, but it is the result of a complex interplay between genetic susceptibility, environmental factors, and the host’s immune response to the gut microbiota. Therapeutic strategies over the past two decades have focused primarily on downregulating the inflammatory response in an attempt to avoid its excessive activation. Because this approach is often insufficient and carries the risk of infectious complications, interest in other microbiota-modulating approaches has grown.

Several studies have shown that gut microbiota differ between healthy individuals and patients with IBD, who have reduced diversity, stability, and resilience, as well as changes in the abundance and function of specific bacterial groups [9]. Moreover, some bacteria, such as adherent-invasive *Escherichia coli* (AIEC) [10], and viral proteins [11] have been identified as potential triggers of intestinal inflammation in IBD.

Given the prominent role of the gut microbiota in the pathogenesis of IBD and other gastrointestinal disorders, attempts to modulate it have been proposed. These include the use of probiotics, prebiotics, synbiotics, antibiotics, dietary interventions, fecal microbiota transplantation (FMT), and bacteriophage therapies. However, drawing general conclusions on the efficacy and safety of these approaches is often difficult due to the variability in the interventions, doses, timing of administration, and the inevitable heterogeneity of host microbiome profiles and intestinal conditions.

To summarize the experiences, perceived relevance, and attitudes of physicians towards microbiome modulation for the treatment of IBD and other gastrointestinal conditions, we conducted an international survey. The survey aimed to evaluate the current practices, preferences, challenges, and expectations of physicians in assessing gut microbiome composition and using antibiotics, probiotics, and FMT. The survey also aimed to identify the factors that influence the decision-making process of physicians and the barriers that limit the implementation of microbiome modulation in clinical practice.

This survey presents a large pool of opinions and experiences in an expanding area of gastroenterology. Its results may provide valuable insights for directing future research and standardization efforts.

## 2. Materials and Methods

We conducted a cross-sectional survey of gastroenterologists and other specialists mostly involved in the management of gastrointestinal disorders and particularly IBD. The survey was administered through a web-based application that was accessible from any device with an internet connection. The questionnaire was drafted by two junior gastroenterologists (T.L.P. and S.V.) and revised by two senior gastroenterologists (L.P.B. and S.D.) with the aim of covering the main issues around microbiota modulation in GI disorders with a particular focus on IBD. Due to timing and attention constraints, an initial cap of 50 questions was set. After rounds of revision and discussion, the final questionnaire consisted of 46 questions that were grouped into six sections, as follows:-The first section included 7 questions on the respondent’s demographics, country of practice, and IBD expertise. The questions asked about the respondent’s age, gender, specialty, years of experience, number of IBD patients seen per month, and setting of practice (academic hospital, secondary centers, etc.);-The second section comprised 2 questions on the respondent’s self-reported knowledge of the gut microbiota and its perceived relevance to the field of IBD;-The third section included 11 questions on the prescription of microbial composition analysis and probiotics. The questions enquired about the frequency, indications, methods, and results of microbial composition analysis performed or prescribed by the respondent, as well as the frequency and outcomes of probiotics prescribed for various GI conditions;-The fourth section included 3 questions on the respondent’s use of antibiotics for IBD, including their frequency, indication and self-reported efficacy on IBD activity;-The fifth section included 7 questions on the respondent’s perceived efficacy, ease of use, and safety of FMT. The questions covered opinions on FMT as a treatment option for IBD patients, the expected benefits and concerns;-A sixth section, of 16 questions, was reserved for respondents who had performed or prescribed FMT for IBD patients. The questions asked about the number, indications, contraindications, route of administration, timing, frequency, follow-up, and outcomes of FMT procedures performed or prescribed by the respondent, as well as the observed efficacy, safety, adverse events, patient satisfaction, and cost-effectiveness of FMT. The full questionnaire is available as Appendix A to this article.

The invitation to participate in the survey was circulated via email using the mailing list of IBDscope^®^, a webinar platform targeted at physicians with an interest in IBD. The survey was open from 17 May to 30 July 2023. In accordance with EU regulations (GDPR), before starting the survey, the respondents were informed about its purpose and scope, the voluntary and anonymous nature of their participation, the use and distribution of the data collected through it, and the estimated time required to complete the questionnaire. Respondents could withdraw consent at any time during the survey, in which case their responses would not have been considered. The respondents were also asked to provide their informed consent before proceeding to answer the survey questions.

## 3. Results

### 3.1. Participant Characteristics

In total, 142 healthcare professionals completed the survey. Geographic origin was quite diverse, with 60 countries represented and the majority of respondents practicing in Europe, followed by South America, the Middle East, East Asia and the Pacific, North America, and Africa (Figure 1).

The average age of respondents was 54 years (SD 13.6), 61% identified as male, and the vast majority were adult gastroenterology specialists (86%); the remaining were pediatric gastroenterologists (4.9%), general surgeons (2.8%), physicians still in training (2.1%), or other specialists (3.5%) (Figure 1).

The majority of participants practiced in academic centers (53%) or tertiary non-academic hospitals (13%), while 11% worked in secondary hospitals and 21% in private practices. Respondents were, for the most part, well-experienced physicians working in the field of IBD for more than 10 years (70%), 18% had 5 to 10 years’ experience, and the remainder were at early career stages. The number of patients with IBD treated was relatively heterogeneous, with roughly one-third treating fewer than 100 patients per year, another third 100 to 500, and the remaining third treating 500 to 1000 (20%) or more than a 1000 (14%) (Figure 1).

### 3.2. Knowledge of Microbiota and Its Perceived Relevance to IBD

Overall, the self-assessment of microbiota expertise was evenly distributed, with the majority of respondents feeling averagely knowledgeable (41%) and 18% and 29% reporting somewhat below- and above-average competence, respectively. Only a few respondents considered themself as not knowledgeable (5%) or experts (7%) in the matter. Despite differences in expertise, there was broad agreement that the gut microbiota are relevant to IBD pathogenesis, with 83% of respondents voting 6 or more on a scale from 0 to 10, where 10 represented maximal relevance.

### 3.3. Microbial Composition Analysis and Probiotics

The survey found that only a small fraction of the respondents (3%) use microbiota composition analysis regularly, while the majority of them (63%) never use this diagnostic tool. The remaining respondents use this technique occasionally (29%) or often (4%) (Figure 2). The respondents who did not use microbiota composition analysis gave various reasons for their decision (multiple answers were possible). The most prevalent was the lack of reference values and variability (intra/interpatient), which was mentioned by 73 respondents. The second most prevalent reason was the lack of reimbursement, reported by 59 respondents, followed by the lack of application to clinical practice, reported by 52 respondents, and difficulty in interpreting the results (40 respondents) (Figure 2).

The survey asked the respondents about their frequency of probiotic prescriptions for patients with Crohn’s disease (CD), ulcerative colitis (UC), and pouchitis. The results showed that 30% of the respondents never prescribed probiotics for CD patients and an additional 31% prescribed them very rarely. Distribution was more balanced for UC, with 19% never prescribing probiotics and 26% more prescribing them very rarely. Prescription patterns differed for pouchitis, for which only 13% never or 8% rarely prescribed them, while a remarkable 54% used probiotics frequently (7–10, where 10 means “always”) (Figure 3).

Apart from the IBD domain, the most common indications for probiotic prescription were post-infection diarrhea, closely followed by antibiotic-associated diarrhea and irritable bowel syndrome (IBS) with predominant diarrhea, all conditions for which probiotics are prescribed frequently (defined as 7–10 on the scale) by one-third to half of participants, whereas probiotics were prescribed the least for constipation-predominant IBS (IBS-C), for which only 27% used them frequently (Figure 3).

### 3.4. Antibiotics for Inflammatory Bowel Disease

Views on antibiotics’ efficacy in inducing and maintaining remission of IBD, excluding fistulas and abscesses, were mixed. A total of 58% of respondents believed antibiotics are useful in inducing disease remission, but only 16% recognized their role for the maintenance of remission. Counterintuitively, a similar proportion (64%) responded that antibiotics could cause IBD flares.

### 3.5. Fecal Microbial Transplant for Inflammatory Bowel Disease

Participants were asked to rate the expected efficacy of FMT on a scale from 1 (no efficacy) to 10 (max efficacy) for five conditions: CD, UC, pouchitis, IBS-C, and IBS-D. The highest perceived efficacy of FMT was for pouchitis, UC, and IBS-D, collecting roughly similar votes. Conversely, IBS-C and CD were thought to benefit the least from FMT. Overall, these results indicate that despite the variation in the perception of gastroenterologists on the efficacy of FMT for different gastrointestinal disorders, FMT is more favorably viewed for disorders with diarrhea as their predominant manifestation and with a strong pathogenic role of bacteria, such as pouchitis (Figure 4).

When asked to compare the efficacy of FMT for IBD with that of approved advanced medications, 61% of respondents believed that FMT was less effective, 21% did not know, 15% thought it was as effective as biologics and novel small molecules, and only 3% thought it was more effective.

Doubts on the duration of FMT’s effects and need for repeated administration were clear, with more than 50% of participants responding that they did not know what the appropriate interval of FMT administration should be, and the remaining 45% roughly equally split on different time intervals (once only, weekly, monthly, every other month, every 6 months, yearly). Instead, there was broad support (74%) for the importance of FMT donors for therapeutic success.

Different opinions emerged on the position of FMT in IBD’s therapeutic algorithm, with 37% of respondents answering they did not know, followed by one-third placing FMT as the last medical option before surgery, 22% after the first medical failure, and 8% proposing it as a first-line treatment.

Of the 142 respondents, 29% would not prescribe FMT for IBD at all, and 12% are not sure. On the other hand, 25% would prescribe it and 34% would consider it for selected cases if it was available and reimbursed. If a clinical trial of FMT for IBD was available at their center, three-quarters of respondents would consider proposing it to their patients, 20% said they were not sure, and 5% would not. Similarly, almost two-thirds thought FMT is safe and 28% were undecided. When asked what their main concerns regarding FMT were, responses focused on safety issues and lack of evidence or guidance for its use.

### 3.6. Experience of Respondents who Performed FMT

Of the 142 total respondents, 37 (26%) performed, prescribed, or administered FMT at least once. Two-thirds used it for UC (67%), 42% for *C. difficile* infection, 28% for IBS, 11% for CD, and 8% for other indications (autism and rheumatoid arthritis—two respondents and one respondent, respectively). The reported efficacy was encouraging, with 47% saying it was very effective and 50% finding it somewhat effective; most importantly, no respondent reported a negative impact of FMT on the disease. Opinions on the ease of preparation and administration were mixed; 64% found it generally easy and the remainder did not. Various routes of administration were used: the majority (64%) administered FMT via lower gastrointestinal endoscopy, rectal enema was used by 22%, 17% used oral formulations (tables, capsules, and enteric-coated capsules), another 17% used nasogastric or duodenal tubes, and finally 11% performed it via upper endoscopy (Figure 5).

In cases of FMT for IBD, in more than half of the responses, FMT was combined with ongoing IBD medication, in almost 20% of responses it was given as monotherapy, and in a few other cases, it was administered together with or preceded by antibiotics, probiotics, prebiotics, and nutritional supplements. A total of 42% of respondents found FMT most efficacious in the induction of remission, and 53% said it worked well for both induction and maintenance. Finally, the main concern/difficulty reported was the unclear evidence supporting FMT (72%), followed by stool preparation (50%) (Figure 6).

## 4. Discussion

The present study provides a comprehensive overview of physicians’ beliefs, attitudes, and experiences concerning gut microbiota modulation through probiotics, antibiotics, and FMT for various intestinal conditions, with a particular focus on IBD. The survey collected responses from healthcare professionals, predominantly gastroenterologists, from diverse geographical regions, offering valuable insights into the current landscape of microbiome-related therapeutic approaches in clinical practice.

Microbiota modulation, particularly through fecal microbial transplant (FMT), remains a relatively underutilized clinical practice, with limited adoption in healthcare settings. Although there is a growing interest in the microbiome and its potential therapeutic applications, it is noteworthy that much of this enthusiasm primarily emanates from the community of research-oriented physicians and scientists. Consequently, it is not surprising that the majority of respondents to our survey were affiliated with academic hospitals, where research and innovation often drive the frontier of medical science and its applications. Despite the interest from and participation of academic centers, it is important to note that only a small minority of the survey respondents had direct experience with fecal microbial transplant (FMT). Of these, some respondents had only performed/administered FMT in experimental settings, for indications not currently included by guidelines recommendations.

One of the main findings of this survey is the consensus among respondents regarding the relevance of the gut microbiota in IBD pathogenesis. This aligns with the growing body of research highlighting the intricate interplay between the gut microbiome, genetic factors, and the immune system in the development and progression of IBD. The acknowledgment of this relationship by healthcare professionals is crucial, as it underscores the importance of microbiome-focused therapeutic strategies. This comes in stark contrast to the much more heterogeneous opinions on how to leverage the microbiome’s role in therapeutic strategies.

To start, the limited utilization of microbial composition analysis in daily practice highlights the lack of standardized reference values, clinical translation of the results, and difficulties in their interpretation. In addition, to a certain degree, costs and ease of reimbursement for extra testing could add to the above-mentioned difficulties in limiting access. These challenges underscore the need for further research to establish clear guidelines for microbial analysis, ideally allowing personalized treatment decisions based on each patient’s unique microbiome profile.

The infrequent use of probiotics, particularly in the management of IBD patients, is notable, though easily explained. Although some strains have been proven to be beneficial, such as *E. coli*
*Nissen* in UC [12,13], the variability in compositions, relatively high costs, and modest evidence supporting their efficacy in multiple settings explain the reluctance to prescribe probiotics for conditions where other treatments are better established. In a similar vein, in the case of IBS with predominant diarrhea or post-antibiotic dysbiosis, where treatment guidance is scarce, the perceived benefit of a probiotic course was greater.

Physicians had conflicting views on the benefits and risks of antibiotics for IBD, reporting at the same time that antibiotics are often useful to induce remission but at the same time are a possible cause of flare. These responses well reflect the shared opinion that microbiota are relevant to IBD and their modulation can be both detrimental and beneficial. Indeed, these views are supported by association studies on antibiotic use and future development of IBD, particularly CD [14], and the seemingly contradictory protective effect of antibiotics on disease flares [15].

The survey revealed varying opinions regarding the perceived efficacy of FMT for different gastrointestinal conditions for which is not an approved treatment. Interestingly, FMT was believed to be most effective for IBS-D and pouchitis and less so for CD and IBS-C. Part of this discrepancy could be explained by the availability of more effective medications for CD, which in turn diminish the perceived benefit of FMT. In line with this, the majority of respondents believed that FMT was less effective than approved advanced medications for IBD. Additional factors such as motility causes of IBS-C are likely to have skewed responses against FMT.

Concerns regarding the duration of FMT’s effects, appropriate administration intervals, and the lack of clear evidence and guidance were prevalent. These uncertainties emphasize the need for rigorous, well-designed clinical trials to establish the safety and long-term efficacy of FMT in various gastrointestinal conditions. Particularly, the long-term implications of FMT are still largely unknown. Two recent real-world studies from the US and from Hong Kong found that the safety profile in the short and medium terms was similar to that observed in the randomized clinical trials. The majority of FMT-related adverse events were self-limiting gastrointestinal symptoms, especially diarrhea and abdominal pain. Some rare, but sometimes serious, cases of infections transmitted through FMT have been described, and to date remain the main safety concern. In the medium term, very few severe adverse events were recorded, and nearly none was attributable to FMT [16,17]. Nevertheless, these cohorts comprise a few hundred patients receiving FMT for various indications through various routes of administration and at different times, and are too heterogeneous to draw definitive conclusions.

The relevance of donor selection to FMT outcomes was widely recognized by the respondents, supporting the importance of standardized donor screening and selection criteria. However, it is worth noting that the association between stool donors and transplant efficacy was mostly observed in the setting of immune-mediated disorders, such as IBD, but not in *C. difficile*, which to date remains the main application of FMT [18]. In recent years, progress on the standardization of FMT procedures has accelerated [19,20,21] and with further studies on the way, the variety of opinions captured in the survey might eventually converge.

Our study has several limitations. Firstly, like any survey, it is subject to sampling bias, with uneven participation from different countries, levels of experience, and settings of practice. Due to these issues, we cannot guarantee that the respondents reflect a reliable sample of the gastroenterologist population. However, the number of participants is relatively large and the provenance of the respondents is quite broad. Moreover, the same mailing list from IBDscope^®^ has been used before for several other surveys supported by respected organizations, including the International Organization for the Study of IBD (IOIBD) [22]. Secondly, knowledge of the microbiota and experience in IBD and FMT were self-reported and would have been impossible to assess or verify independently. The topics covered were the results of a difficult compromise between the length of the questionnaire and relevance to clinical practice. This led us to prioritize the bacterial microbiome at the expense of other microorganisms, primarily viruses and fungi, which, despite gaining recognition for their role, are more distant from clinical applications [23,24,25]. Similarly, we chose to address *C. difficile* infection relatively less, as evidence in support of FMT and microbiota modulation is more robust and individual experience proportionally less insightful [26,27,28]. Finally, the relatively small number of respondents with direct experience of FMT limits the generalizability of that part of the results.

## 5. Conclusions

In conclusion, this international survey of physicians provides valuable insights into the beliefs, attitudes, and experiences related to gut microbiota modulation in the management of gastrointestinal disorders, particularly IBD. While there is consensus on the relevance of the gut microbiota in IBD, the survey highlights several challenges and knowledge gaps in the clinical application of microbiome-based therapies. These findings underscore the need for further research, standardization of techniques, and evidence-based guidelines to optimize the use of probiotics, antibiotics, and FMT in clinical practice. As our understanding of the gut microbiome continues to evolve, these insights will contribute to more informed and personalized approaches to managing gastrointestinal conditions.

## Figures and Tables

**Figure 1 microorganisms-11-02806-f001:**
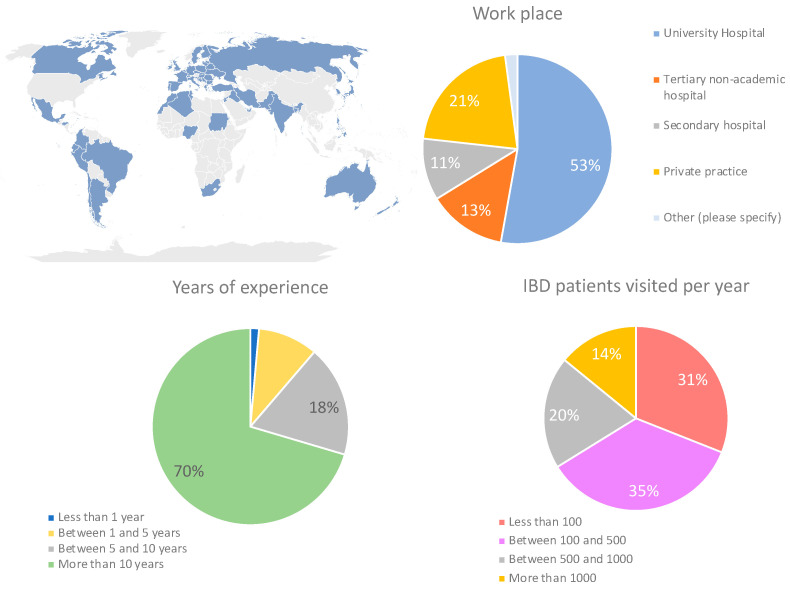
Country of work, practice setting, and experience of survey respondents.

**Figure 2 microorganisms-11-02806-f002:**
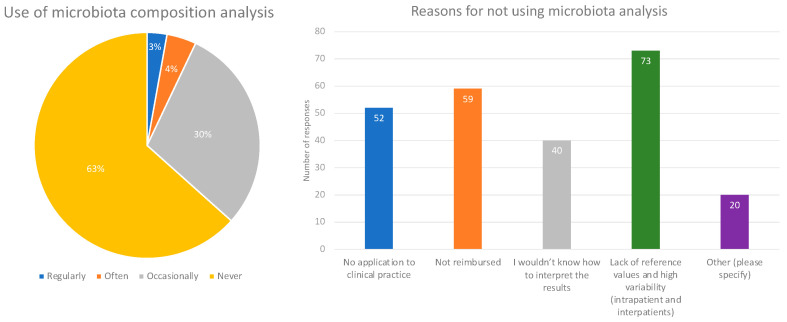
Use of microbiota composition analysis and reasons for non-use.

**Figure 3 microorganisms-11-02806-f003:**
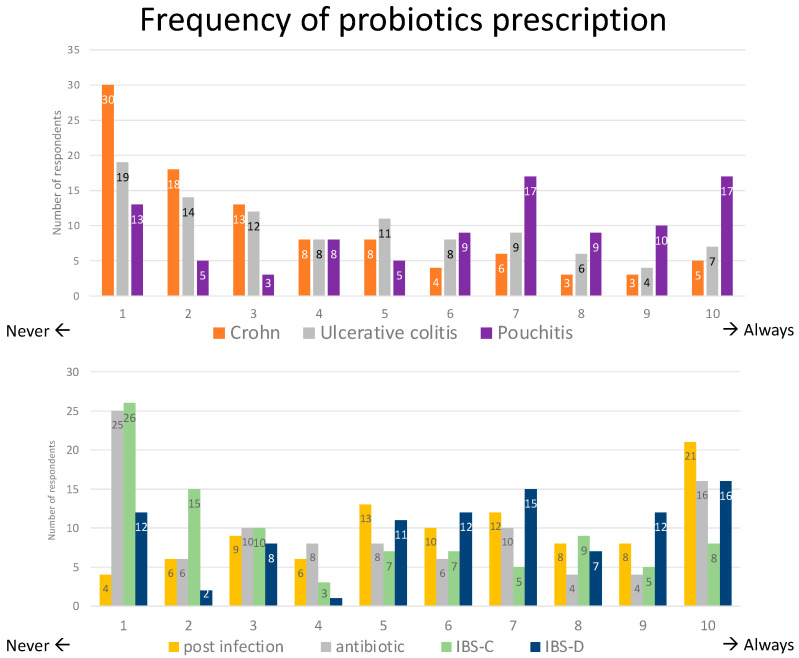
Frequency of probiotic use for different gastrointestinal conditions.

**Figure 4 microorganisms-11-02806-f004:**
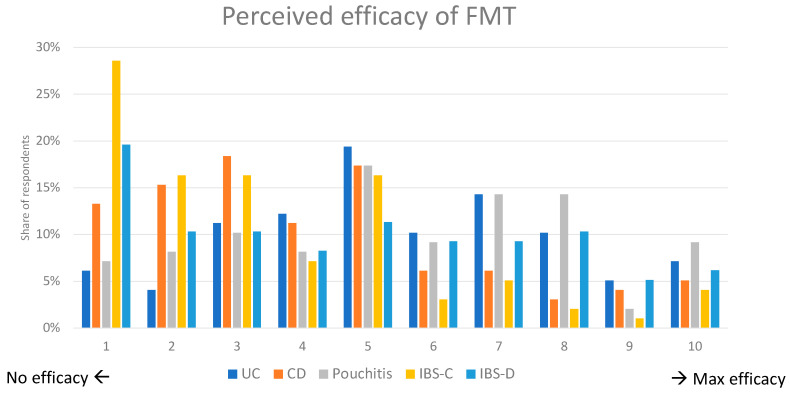
Perceived efficacy of fecal microbial transplant (FMT) for different gastrointestinal conditions.

**Figure 5 microorganisms-11-02806-f005:**
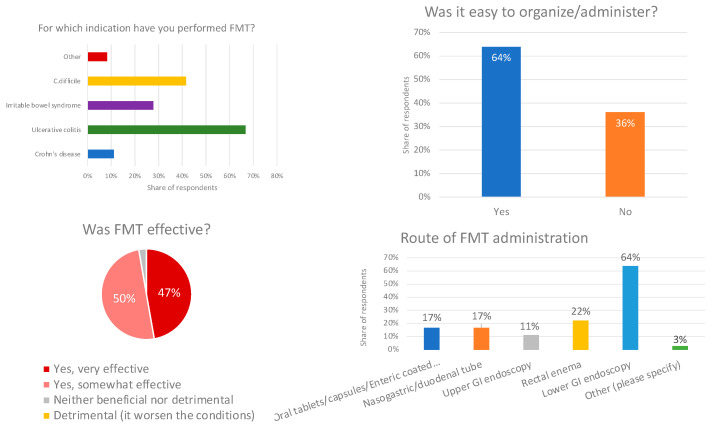
Direct experiences of respondents on use of fecal microbial transplant, indications, observed efficacy, and ease and route of administration.

**Figure 6 microorganisms-11-02806-f006:**
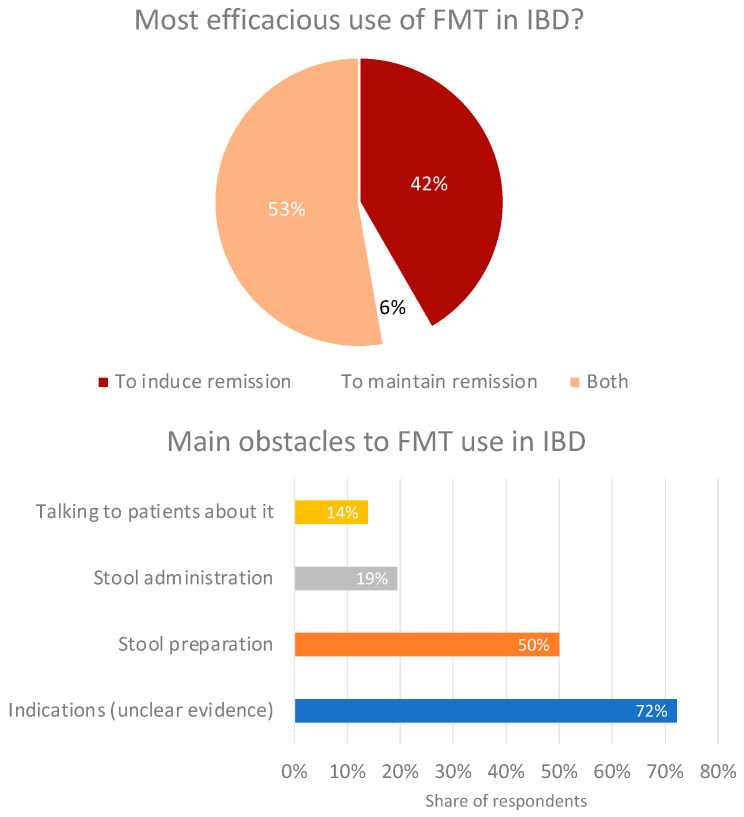
Perceived best use of fecal microbial transplant in IBD (top) and obstacles to its adoption (bottom).

## Data Availability

Anonymized data of the survey will be made available at reasonable request to the corresponding author.

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
