# Peer review of "Efficacy, Safety, and Concerns on Microbiota Modulation, Antibiotics, Probiotics, and Fecal Microbial Transplant for Inflammatory Bowel Disease and Other Gastrointestinal Conditions: Results from an International Survey"

_microorganisms, 2023, doi:10.3390/microorganisms11112806_

Round 1

Reviewer 1 Report

Comments and Suggestions for Authors

Informative manuscript presenting data of a survey among gastroenterologists on the current knowledge and clinical practice of microbiota modulation (probiotics, antibiotics and fecal microbiota transplantation) in GI disorders. 

Introduction: excellent introduction, very informative, well written. 

1)   Please modify last sentence: “Its results may provide valuable insights for improving the management of patients with gastrointestinal disorders and direct future research”.  Knowledge gaps are indentified and these may direct future research. Improving management of patients with GI disorders is not an aim of your study. 

Methods: 

2) about the questions : how was the process of selection of questions for the questionnaire? Have experts given input?  Do these items reflect the main relevant and/or critical issues around microbiota modulation in GI disorders?  

3) Is the doctors subset a representation of the general worldwide gastroenterologists  population? 

4) were the levels of microbiota knowledge predefined, or was the level fully self-perceived?

Results: 

Well presented and concise.

5)   FMT: why was therapy resistant Clostridium diff. infection not taken into account?

6)    Fig 5 upper left corner: For which indication have you performed FMT : 50% answered: other. Please specify. It will be interesting to add these data. 

7)   obstacles to use FMT in IBD: have risks and lack of knowledge of long term side effects and consequenses been mentioned ? 

Discussion:

Informative, well written. 

Conclusions are in line with the findings and exactly pinpoint the knowledge gaps.  

Author Response

We thank the reviewer for the many precious suggestions we have fully implemented. Please find below a point-by-point answer in red.

Informative manuscript presenting data of a survey among gastroenterologists on the current knowledge and clinical practice of microbiota modulation (probiotics, antibiotics and fecal microbiota transplantation) in GI disorders. 

Introduction: excellent introduction, very informative, well written. 

1)   Please modify last sentence: “Its results may provide valuable insights for improving the management of patients with gastrointestinal disorders and direct future research”.  Knowledge gaps are indentified and these may direct future research. Improving management of patients with GI disorders is not an aim of your study. 

We thank the reviewer for the thoughtful comment, we have corrected the sentence accordingly. (Lines 80-82)

Methods: 

2) about the questions : how was the process of selection of questions for the questionnaire? Have experts given input?  Do these items reflect the main relevant and/or critical issues around microbiota modulation in GI disorders?  

Thanks for pointing this out. We have now clarified in the methods that the questionnaire was drafted by two junior gastroenterologists and then revised by two seniors. Due  attention and timing/compliance constraints we initially limited the length of the survey to 50 questions, then short listed it to 46, with the intent to cover the most relevant issues on microbiota modulation. (Lines 88-92)

3) Is the doctors subset a representation of the general worldwide gastroenterologists  population? 

As for any survey, this is a critical point. Although the mailing list of IBD scope reaches a very broad audience, and has been used before for many other surveys, despite our best efforts, we cannot guarantee that the respondents represent an accurate subset of the world gastroenterologists population. Participation in this sort of initiatives tends to be skewed in favour of academic/tertiary centres mainly from Europe and South America. Nevertheless, because a perfect representation of the gastroenterologists population is nearly impossible to provide we still believe the results are informative and worth sharing. Due to this known limitations we reported in the detail the setting of practice (academic, tertiary, secondary etc), the country of practice, and the level of expertise in the results section. Moreover, in light of this comment we expanded the part of the discussion acknowledging this limitation. (Lines 312-317)

4) were the levels of microbiota knowledge predefined, or was the level fully self-perceived?

Microbiota knowledge was self-reported, in absence of any standard curricula unfortunately this was the only assessment possible. We clarified it in the text  (317-318)

Results: 

Well presented and concise.

5)   FMT: why was therapy resistant Clostridium diff. infection not taken into account?

Like for any survey the number of questions is limited and topics need to be prioritised. We intentionally chose to dedicate less attention to management of C.difficile through microbiota modulation because stronger evidence and guidance exist and therefore individual experience and perception is proportionally are less relevant. We added a few lines in the discussion to clarify it. (328-330)

6)    Fig 5 upper left corner: For which indication have you performed FMT : 50% answered: other. Please specify. It will be interesting to add these data. 

We thank the reviewer for his/her suggestion. We changed the figure reporting also the share of respondents that did FMT for C.diff, and specified in the body of the article  that some respondents used FMT for other indications (rheumatoid arthritis 1, and autism 2). Lines 233-234

7)   obstacles to use FMT in IBD: have risks and lack of knowledge of long term side effects and consequenses been mentioned ? 

This is certainly an important point. After the reviewer suggestion we expanded the discussion commenting on it. Lines 307-308

Discussion:

Informative, well written. 

Conclusions are in line with the findings and exactly pinpoint the knowledge gaps.  

Reviewer 2 Report

Comments and Suggestions for Authors

The article presents a very interesting survey on the specialists' point upon the use of microbiota modulation in inflammatory bowel diseases. The subject is of novelty and very interesting for the readers. The authors found that, despite acknowledge the importance od microbiota in these diseases, very few subjects use microbiota testing and modulation in their practice.

One possible explanation, may be, in my opinion, the limitted posibilities of testing due to economic reasos and lack of technical equipment.

Some minor English errors should be corrected.

Comments on the Quality of English Language

Some minor English language should be corrected.

Author Response

We thank the reviewer for his/her positive and thoughtful comments. We are glad that the article was appreciated.

As rightly pointed out, we have added a comment on the cost of microbiota analysis and how this could impact access. Line 275-276

Reviewer 3 Report

Comments and Suggestions for Authors

Dear Editor,

I carefully read the manuscript "Efficacy, safety, and concerns on microbiota modulation, antibiotics, probiotics and fecal microbial transplant for inflammatory bowel disease and other gastrointestinal conditions: results from an international survey".

My comments and suggestions for the authors are the following:

 - 142 health care professionals completed the survey and 60 countries were represented. This means that approximately 2 people per country participated in the survey. Is not that a bit little? Why so few health care professionals responded to the survey, according to the authors' opinion? This issue should be addressed in the manuscript.

 - Pag. 2: The authors misused "gender" in place of "sex". Sex refers to “the different biological and physiological characteristics of males and females, such as reproductive organs, chromosomes, hormones, etc.” Gender refers to "the socially constructed characteristics of women and men – such as norms, roles and relationships of and between groups of women and men.

 - Study's limitations should be further and more deeply discussed.

 - Which was the ethics committee that approved the study? More information should be included in the manuscript.

Author Response

  • We aimed to reach the largest possible number of respondents. The invite was circulated through the mailing list of IBDscope, a webinar platform dedicated to IBD. Inevitably the audience of IBDscope is not evenly distributed across countries and its network is strongest in Europe and South America. Some countries contributed with one respondent only, other with multiple. We believe that the number of respondents is large enough to draw meaningful conclusions, while geographic-based subanalysis was not conducted nor planned.
  • On page 2 we describe the variables collected from the respondents because we are interested in the characteristics of the respondents because this could affect their opinions we chose to let them self-identify in whichever gender best reflects them, instead of distinguishing in biological sexes which is less accurate in this context.
  • The limitation section of the discussion has been extensively expanded.
  • No ethics approval was necessary. All participants before starting the survey agreed to the terms and conditions. These, in accordance with EU's regulation (GDPR) detailed the purposes, nature, use, and distribution of the data collected, including the present publication.

Round 2

Reviewer 3 Report

Comments and Suggestions for Authors

Dear Editor,

I carefully read the revised version of the manuscript and authors' reply to my comments. Unfortunately, I still have serious concerns as regards the publication of the article.

It is very interesting that the authors collected the variable "gender" in their study. However, they should have also collected the "sex" variable (biologically, both these variables are important and it is not true -as they wrongly claim- that in this context it is more critical to know the gender of the respodants than their sex).

As regards the Ethical Approval... the Italian Regulatory Agency (AIFA) requires that also observational studies are previously approved by an ethics committee. If (as I believe!) the Principal Investigator of this study is based in Italy, She/He must submit to the competent regulator, of course.

Author Response

Response:

  1. We appreciate the reviewer’s concerns regarding the missing data on respondents’ sex, however we believe this is a marginal aspect that does not impact the quality of the data nor the reliability of the conclusions. We chose to ask participants their gender, instead of their sex, first to be more inclusive towards anyone who’s gender identity does not match his/her sex; secondly because opinions (our data of interest) are more shaped by social/cultural factors better reflected by gender identity rather than mere biological classification. In any case, no sub-analysis included the gender variable, therefore no result or conclusion would have differed had we asked for sex instead of gender. To collected both sex and gender would have been beyond the scope or need of the study.
  2. AIFA regulation applies to clinical observational studies of patients. We conducted a survey of physicians, without any patient involvement, hence no ethical approval was needed. We can reassure the reviewer, as it’s clearly stated in the manuscript, that all participants agreed the terms and conditions of the survey that detailed the type, use and publication of the data collected.